# Implementation of a multicomponent family support intervention in adult intensive care units: study protocol for an embedded mixed-methods multiple case study (FICUS implementation study)

Saskia Oesch  ,[1,2] Lotte Verweij,[1,2] Lauren Clack,[1,3] Tracy Finch,[4] Marco Riguzzi,[1,2] Rahel Naef[1,2]

¹Institute for Implementation Science in Health Care, University of Zurich Faculty of Medicine, Zurich, Switzerland
²Center of Clinical Nursing Science, University Hospital Zurich, Zurich, Switzerland
³Division of Infectious Diseases and Hospital Epidemiology, University Hospital Zurich, Zurich, Switzerland
⁴Nursing, Midwifery and Health, Northumbria University, Newcastle upon Tyne, UK

**Correspondence to**
Saskia Oesch;
saskia.oesch2@uzh.ch

## ABSTRACT

**Background** The implementation of complex interventions is considered challenging, particularly in multi-site clinical trials and dynamic clinical settings. This study protocol is part of the family intensive care units (FICUS) hybrid effectiveness-implementation study. It aims to understand the integration of a multicomponent family support intervention in the real-world context of adult intensive care units (ICUs). Specifically, the study will assess implementation processes and outcomes of the study intervention, including fidelity, and will enable explanation of the clinical effectiveness outcomes of the trial.

**Methods and analysis** This mixed-methods multiple case study is guided by two implementation theories, the Normalisation Process Theory and the Consolidated Framework for Implementation Research. Participants are key clinical partners and healthcare professionals of eight ICUs allocated to the intervention group of the FICUS trial in the German-speaking part of Switzerland. Data will be collected at four timepoints over the 18-month active implementation and delivery phase using qualitative (small group interviews, observation, focus group interviews) and quantitative data collection methods (surveys, logs). Descriptive statistics and parametric and non-parametric tests will be used according to data distribution to analyse within and between cluster differences, similarities and factors associated with fidelity and the level of integration over time. Qualitative data will be analysed using a pragmatic rapid analysis approach and content analysis.

**Ethics and dissemination** Ethics approval was obtained from the Cantonal Ethics Committee of Zurich BASEC ID 2021-02300 (8 February 2022). Study findings will provide insights into implementation and its contribution to intervention outcomes, enabling understanding of the usefulness of applied implementation strategies and highlighting main barriers that need to be addressed for scaling the intervention to other healthcare contexts. Findings will be disseminated in peer-reviewed journals and conferences.

**Protocol registration number** Open science framework (OSF) https://osf.io/8t2ud Registered on 21 December 2022.

## STRENGTHS AND LIMITATIONS OF THIS STUDY

⇒ The use of hybrid study designs to understand intervention and implementation outcomes is key in complex intervention health research.
⇒ The proposed study draws on established implementation science theories and methods to investigate the integration and delivery of a complex family support intervention (FSI) in a highly dynamic clinical environment.
⇒ A strength of this mixed-methods study is a balanced use of rigorous and pragmatic data collection and analysis strategies to inform ongoing implementation while also generating generalisable knowledge on implementation strategies and determinants around complex FSIs.
⇒ Given the anticipated small sample sizes per case, potential limitations will be a small or incomplete quantitative data set due to staff turnover or low response rate. Another potential limitation of the study might be the data collection burden on participants.
⇒ The use of several data sources and the combination of qualitative and quantitative data will enable a comprehensive understanding of the phenomenon under study.

## INTRODUCTION

Up to 60% of family members of critically ill patients in intensive care units (ICUs) experience high levels of stress and uncertainty, which may lead to negative mental health outcomes.[1–3] Families require consistent involvement, communication and support, however, there is a lack of research on the clinical effectiveness and successful implementation of family-focused care in ICUs.[3] To better support family health and to close this know-do gap in the area of family-focused ICU care, the family intensive care units (FICUS) study was launched to investigate the effectiveness and implementation

of a nurse-led, interprofessional delivered family support intervention (FSI) in Swiss ICUs.[4] Successful implementation of the three FSI components, namely family engagement, support and communication by a family nurse and interprofessional team along the patient pathway is expected to increase the likelihood of clinical effectiveness, and will be critically influenced by the context into which the intervention is introduced, particularly as part of a multisite clinical trial.[5–8]

Specific implementation guidance for integrating FSIs in critical healthcare settings is still scarce. Previously published research focusing on family nursing implementation have mainly focused on promoting nurses' skills development and open attitudes toward family-focused care.[9–13] Main barriers to implement family-focused care in ICUs have been described as lack of understanding of what needs to be done to achieve family-centred care (eg, lack of support from fellow nurses), organisational barriers (eg, daily workload), individual barriers (eg, healthcare professionals' (HPs') attitudes) and barriers related to interprofessional care (eg, communication barriers). In contrast, leadership support, clear and accurate communication, and engagement of the healthcare team are described as the most important facilitators.[14–24] However, intervention research around FSIs in ICUs has not investigated their implementation,[12 25 26] except for one ongoing trial, which will explore implementation processes and outcomes.[27]

The implementation of complex interventions, such as FSIs, is challenging as they contain several independently and interdependently interacting components at the individual behavioural, interprofessional team and organisational level.[6–8 28–30] Implementation of new interventions involves changes at the individual behavioural, team and organisational level[31] and several contextual determinants that influence successful integration may appear.[32] A growing body of research focuses on the development of theories and frameworks to optimise the design and implementation of complex interventions,[7 8 33 34] but implementation success remains still widely variable.[35] FSIs pose greater challenges for implementation evaluation and cannot rely on pre-established cause and effect specifications.[36] As recommended within implementation science and complex intervention research, the focus should not only lie on the clinical effectiveness of a given intervention, but on the conditions needed to achieve the intervention's reach and impact in the real-world context.[7 8] Randomised controlled trials are important in establishing intervention effectiveness, however they focus solely on clinical health or service outcomes.[37 38] Even if implementation outcomes, such as fidelity, serve as intermediate outcomes in studies of intervention effectiveness,[39] in-depth exploration of implementation, particularly in the field of FSIs, has received scant attention.[12 40]

A focus on implementation is needed to learn about successful integration processes of FSIs in critical care settings. This paper reports an implementation study protocol for investigating the implementation process and outcomes of a multicomponent FSI alongside a cluster-randomised clinical trial (FICUS),[4] which examines the clinical effectiveness.

## METHODS
### Design
The current study is part of the FICUS hybrid effectiveness-implementation study and is conducted between June 2022 and January 2024. To investigate the implementation processes and outcomes of a complex phenomenon, that is the FSI in its 'real-world' context,[41] we will use an embedded multiple case study—mixed methods convergent design,[42 43] in which each of the participating ICUs is conceptualised as a 'case'.[44] Two research questions will be investigated:

► What influence does the integration process of the FSI in eight ICUs have on the level of integration, mechanisms of impact and the implementation outcomes defined as fidelity, feasibility, acceptability, appropriateness and sustainability?
► Which contextual determinants (barriers and facilitators) influence the integration process and explain its impact on the intervention outcomes?

### Patient and public involvement
Within the FICUS study, an advisory board was established with a patient expert, three family members and a patient with life experience in critical care. The advisory board began its work with the research team during the application phase of the FICUS study and continues to actively participate in the study.

### Study overview
Qualitative and quantitative data will be collected at four timepoints figure 1. First, data will be collected as part of a formative process evaluation during the adaptive implementation process at three timepoints over 12 months (T1–T3). Rapid, primarily qualitative data collection procedures, such as small group interviews, observation and case conferences will be applied,[45–47] coupled with structured self-assessment tools, and integrated into a series of meetings and activities with key clinical partners (KP) that serve to support implementation and to ensure quality of intervention delivery. A summative evaluation will be held with KP and HP after the conclusion of the implementation/delivery phase (T4) to assess all study endpoints, using quantitative and qualitative assessment methods, such as focus group interviews and surveys.

### Guiding conceptual frameworks and theories
The Normalisation Process Theory (NPT)[48 49] and the Consolidated Framework for Implementation Research (CFIR)[50] will guide the evaluation of the implementation. While NPT explains changes in the way people think about and act to integrate a new intervention, CFIR categorises and describes contextual determinants that influence

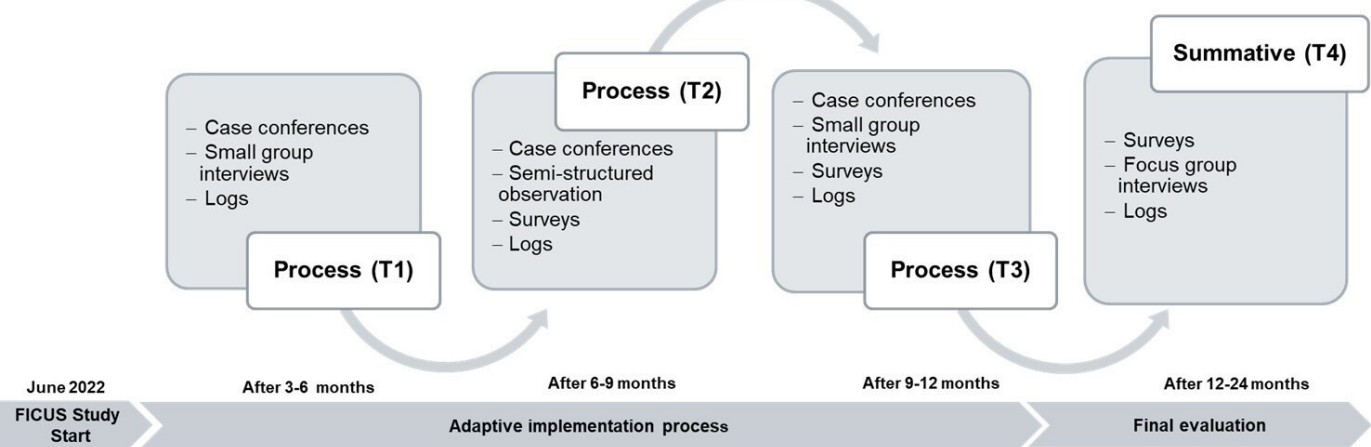

**Figure 1** FICUS implementation study overview.

implementation.[51] NPT and CFIR will be combined and integrated in all stages of the research process, including conceptualisation, data collection, analysis and dissemination. Table 1 provides an overview and description of NPT and CFIR in the implementation study.

We developed an Implementation Research Logic Model drawing on CFIR,[50] NPT[31] and Proctor's Conceptual Framework for Implementation Research[39] to develop a specific implementation theory around the FSI that outlines contextual determinants implementation mechanisms of impact, and outcomes (figure 2). This approach aligns with the realist evaluation paradigm because it attends to context, mechanisms and outcomes of a complex intervention implementation, but uses

| Table 1 | Selected implementation science frameworks and theories used within the FICUS implementation study | |
|---|---|---|
| **Framework/theory** | **Type and description** | **Relevance to the FICUS implementation study** |
| NPT | ▶ The NPT is a middle-range sociological theory that conceptualises implementation, embedding and integration of complex interventions in healthcare settings.[49]<br>▶ It focuses on the work required to embed and normalise an intervention in routine practice. NPT consists of 4 key constructs (coherence, cognitive participation, collective action and reflexive monitoring) and 16 subconstructs.[80] | ▶ NPT provides a flexible framework to evaluate integration and embedding of the FSI.<br>▶ NPT will help to first, better understand the way implementation processes and context shape each other, and second, to explore the collaborative work people do to make sense of the FSI and to integrate it in ICU clinicians' daily routine.[63]<br>▶ NPT will be used to investigate and to describe the integration and normalisation of the FSI in ICU teams. |
| CFIR* | ▶ The CFIR is a meta-theoretical determinant framework which provides a pragmatic structure to guide process evaluations.<br>▶ CFIR provides an overarching typology to promote implementation theory development and verification about what works where and why across multiple contexts.[50]<br>▶ It includes five domains (inner setting, outer setting, intervention characteristics, characteristics of individuals involved and processes of implementation).[50]<br>▶ Within the five domains are 37 constructs that can each act as a barrier and/or facilitator to the implementation of an intervention. | ▶ The CFIR framework includes a wide range of constructs that relate not only to individual characteristics, but also to structural characteristics of the healthcare system, as well as on characteristics of the intervention itself.<br>▶ CFIR provides a degree of flexibility where constructs can be selected that are the most relevant to understand individual and system barriers and facilitators associated with the implementation[50] and to assess key contextual determinants to consider within the outer setting and the intervention characteristics domains, which are outside of the NPT scope.[51]<br>▶ CFIR will be used to describe potential influencing determinants (barriers and facilitators) during implementation. |

*We will use the first version of CFIR[50] because our contextual analysis to adjust the implementation was conducted before the publication of CFIR version 2.0.[81]

CFIR, Consolidated Framework for Implementation Research; FICUS, family intensive care units; FSI, family support intervention; ICU, intensive care unit; NPT, Normalisation Process Theory.

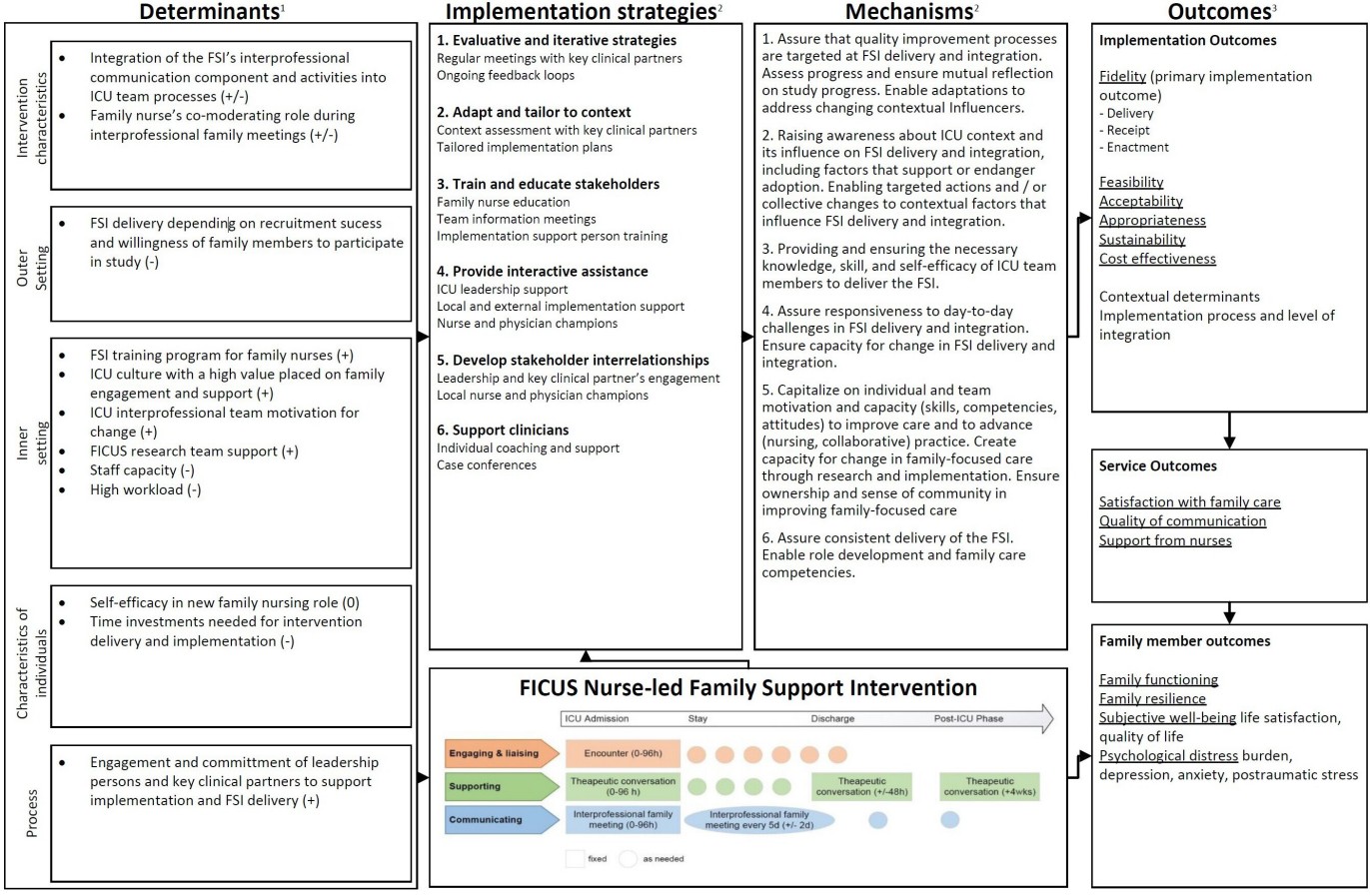

**Figure 2** Implementation Research Logic Model. CFIR, Consolidated Framework for Implementation Research; FICUS, family intensive care units; FSI, family support intervention; ICU, intensive care unit; NPI, Normalisation Process Theory. [1]Identified determinants (influencers) during context assessment before FICUS study start based on CFIR (0 neutral, + positive, - negativ, +/- positiv and negativ) (Damschroder et al., 2009), [2]Implementation strategies based on Naef et al. (2022), [3] Implementation strategy - mechanims based on NPT (May et al., 2009), [4] Measured outcomes according to the conceptualization of Proctor et al. (2009).

specific theories and frameworks from implementation science to do so.[52–54]

### Setting
The FSI will be implemented in eight of sixteen ICUs participating in the FICUS trial, which have been randomly assigned to the intervention arm.[4] All ICUs are certified by the Swiss Society for Intensive Medicine; two are situated in a university affiliated, and six in a cantonal or teaching hospital.

### Participants and sampling
Data will be collected from KP and HP by a doctoral student and a senior researcher. The senior researcher is trained and skilled in qualitative data collection and will train and supervise the doctoral student in doing so. KP are local clinical and management staff involved in the implementation and delivery of the FSI (n=45) and will include local implementers (LI) who support the implementation process, trained family nurses (FN), who are registered nurses with a certification in ICU nursing or equivalent, who are trained in family systems nursing and deliver the FSI, as well as a nurse and physician team

leaders and/or their deputy. Quantitative and qualitative data will be gathered from all KP during the process evaluation.

The second participant group will include HP working on the participating ICUs (approximately n≈750). Quantitative data for the summative evaluation will be gathered from all HP with an anticipated response rate of 50%–70% using a comprehensive sampling strategy. To increase response rates, 3-weekly reminder emails will be sent twice. Qualitative data will be gathered from a subsample of HP (n=80–100) using a purposive sampling strategy. Inclusion criteria will be: ICU physicians or nurses working clinically on study units, having an active or influential role in the implementation/delivery of the FSI and/or having either knowledge/experience with the FSI and/or a clinical leadership responsibility, having worked at the ICU for at least 6 months.

### Study endpoints
The study endpoints pertain to three conceptual dimensions: (1) process indicators, (2) implementation outcomes and (3) contextual determinants. Table 2

**Table 2** Summary study endpoints, operationalisation and data collection timepoints

| Study endpoints | Operationalisation | Data collection timepoint | | | |
|---|---|---|---|---|---|
| | | T1 | T2 | T3 | T4 |
| **Implementation process** | | | | | |
| Process indicators | Small group interviews* | x | | x | |
| | NoMAD*†‡ | | | x | x |
| | Implementation rating tool* | x | | x | |
| **Implementation outcomes** | | | | | |
| **Fidelity: delivery** | | | | | |
| Relative frequency | Intervention fidelity log*§ | x | x | x | x |
| Absolute dose | Intervention fidelity log*§ | x | x | x | x |
| Consistency | Intervention fidelity log*§ | x | x | x | x |
| Availability | Intervention capacity assessment* | x | | | |
| | Focus group interviews† | | | | x |
| **Fidelity: receipt** | | | | | |
| Attendance at case conferences | Case conferences*¶ | x | x | x | x |
| Comprehension and skills | FNPS*** | x | x | | x |
| | Focus group interviews† | | | | x |
| **Fidelity: enactment** | | | | | |
| Engagement and performance | Fidelity Self-Rating Tool* | | x | | x |
| | Semi-structured observation* | | x | | |
| | Focus group interviews† | | | | x |
| Feasibility | FIM*††† | x | | x | x |
| | Focus group interviews† | | | | x |
| Acceptability | AIM*†‡‡ | x | | x | x |
| | Focus group interviews | | | | x |
| Appropriateness | IAM*†§§ | x | | x | x |
| | Focus group interviews† | | | | x |
| Sustainability/sustainment | Sustainability implementation survey¶¶ | | | | x |
| | Focus group interviews† | | | | x |
| Contextual determinants | Small group interviews* | x | | x | |
| | Focus group interviews† | | | | x |

*Part of the process evaluation.
†Part of the summative evaluation.
‡German Version of the Normalisation Measure Development questionnaire (NoMAD).[68]
§Case report forms embedded in REDCap[65] and completed throughout the study by family nurses.
¶Monthly case conferences throughout the 18-month active implementation and delivery phase.
**German version of the Family Nursing Practice Scale (FNPS).[66]
††German version of the Feasibility Intervention Measure (FIM).[67]
‡‡German version of the Acceptability Intervention Measure (AIM).[67]
§§German version of the Intervention Appropriateness Measure.[67]
¶¶Data collection occurs 6 and 12 months after active implementation and intervention delivery/family intensive care units study end.
CFIR, Consolidated Framework for Implementation Research; FN, family nurses; HP, healthcare professionals; ICU, intensive care unit; KP, key clinical partners; LI, local implementers; LoS, length of stay; NPT, Normalisation Process Theory.

provides a summary of the study endpoints, operationalisation and data collection timepoints. A more detailed description of the study endpoints, definitions, data source, operationalisation and data collection timepoints is available as online supplemental material.

**Process indicators**

The implementation process and integration mechanisms will be investigated to gain an in-depth understanding of the implementation from the perspective of KP. It is of particular interest how the FSI becomes (or

fails to become) integrated into interprofessional care delivery to families and a normalised ICU care process.[48] Indicators will include self-perceived integration of the intervention in the interprofessional ICU team and the overall level of integration in ICU processes.

## Implementation outcomes

We will investigate the implementation outcomes of fidelity, feasibility, acceptability, appropriateness and sustainability.[39] As fidelity is vital to evaluate our intervention and implementation theory for predicting effectiveness outcomes,[55 56] we chose it as the primary outcome. Fidelity is defined according to the Conceptual Framework for Implementation Fidelity,[57 58] and has been operationalised within the three domains of delivery, receipt and enactment as outlined by Bellg and colleagues.[59]

We will investigate feasibility, appropriateness and acceptability because these outcomes play a critical role in implementation research and serve as indicators of the effect of the implementation process and as prerequisites for achieving the desired service and family outcomes.[60] To understand how the implementation develops over time and how to best support long-term delivery, we will investigate sustainability as well as post implementation sustainment outcomes.[61]

## Contextual determinants

Contextual determinants at the level of individuals, the team, the organisation and the implementation itself, namely barriers and facilitators that hinder or support successful implementation, will be assessed from a clinician's perspective guided by the CFIR[50] and NPT.[48]

## Qualitative data collection
### Small group interview

Small group interviews refer to a semi-structured discussion with a small group of KP (four to five participants) to facilitate effective discussion on the implementation process and ensure that each KP is adequately heard. This method will allow for gathering information, exploring opinions and understanding perspectives.[62] A total of 16 small group interviews will be held with KP as part of the process evaluation to assess contextual determinants (barriers and facilitators) and processes that influence the implementation. An interview guide, which is based on NPT will be used to explore participants' understanding and perceptions of the intervention delivery, their team progress to achieve integration, and the barriers and facilitators they currently face.[63] Small group interviews will include questions such as: was the defined implementation strategy useful, and do any adjustments or additional measures need to be taken?; what are the reasons that currently hinder the implementation of the FSI, and what have been the biggest challenges so far and what are some ideas on how to overcome these barriers in the next phase of implementation?; is the FSI capable of addressing the needs of the families and patients and how well does the FSI fit into current practices and processes,

and does it require any adjustments? All interviews will be held face-to-face by two researchers at the respective ICU, audio-recorded and expected to last 60–90 min. Immediately after each interview, key topics will be summarised in a protocol and provided to the KP for validation. This form of member checking will give them the ability to correct errors and wrong interpretations and to create trustworthiness.[43]

### Focus group interview

Focus group interviews differ from small group interviews as they are more structured and conducted with a larger number of participants (8–10 participants) to obtain a comprehensive picture of the FSI implementation.[62] A total of 10 focus group interviews will be held as part of the summative evaluation with KP and HP as it is a useful way of collecting qualitative data as social interactions, shared values, cultures and practices which are of particular interest.[64]

First, one focus group interview will be held on each ICU with a purposive sample of 8–10 KS and HP, which represent different professional groups, roles and levels of involvement in the intervention and implementation. A semi-structured interview guide will be used including open-ended questions focusing on: how do ICU staff describe the implementation and feasibility of the FSI on their ICU, what team processes and implementation activities took place during the implementation of the FSI?; what factors were supportive or inhibiting and how did the team perceive the role of FN's and the family care pathway?; and what impact (benefits, unexpected consequences) did they experience in their daily work, interprofessional and with patients and their families?

In addition, two focus group interviews will be held with 8–10 FN using an interview guide that includes open-ended questions on FN's experiences and perspectives on their role adoption, team-based intervention delivery and clinical work with families, experiences with FSI implementation in the study context, observed benefits for themselves, the team and the families, as well factors that promoted or hindered intervention delivery and integration into ICU care.

All focus group interviews will be held face-to-face by two researchers and will take place on site at the respective ICU or an accessible location, in a quiet room to avoid interruptions. Interviews will be digitally recorded, transcribed and field notes will be taken. The interviews and the field notes will be transcribed, checked for accuracy, anonymised and entered into MAXQDA for analysis.

## Observation of intervention delivery

A total of eight semi-structured observations of intervention delivery by trained FN, that is, one per ICU, will be conducted to assess intervention fidelity and the establishment of structures for the FSI using a semi-structured fidelity audit observation tool. The tool bases on the intervention manual and was developed by the research team. It consists of four items on available structures for

the FICUS study and 59 FSI activities corresponding to the three main components (1) engaging and liaising, (2) supporting and (3) communicating. Activities will be rated according to their performance by the FN (performed, partially performed, not performed, not applicable). In addition, the timepoint of the conversation (eg, admission) and the type of conversation (eg, on-site, via telephone) will be documented. An additional text box at the end of the tool will allow to document observations about the ICU context. The tool was pretested once by one researcher. All observations will be conducted by a doctoral student or a researcher, which are both trained in family systems nursing and are expected to last 2–4 hours each.

### Case conference

Monthly case conferences with FNs will be held as moderated group discussions to ensure quality of FSI delivery and to support consistency and adherence to the FSI activities. During case conferences, a specific family case will be presented by FN's and FN's will have the opportunity to discuss, to share their experiences and to reflect on the presented case. Based on the discussions, field notes will be taken by a doctoral student and FNs' participation rate will be captured, as it is an indicator for fidelity receipt. Participation of the FN will be documented in a structured participation form. All case conferences will be held online by two senior researchers trained in family systems nursing and will last 60 min.

### Quantitative measures

#### Fidelity Self-Rating Tool

To capture quality and consistency of intervention performance (fidelity enactment), FN's appraisal of self-perceived engagement and performance in FSI delivery will be assessed two times using the Fidelity Self-Rating Tool (Fidelity-SRT). The tool is based on the intervention manual and was developed by the FICUS research team. It includes three subscales according to the intervention components: (1) engaging and liaising (15 items), (2) supporting (27 items) and (3) communicating (15 items). All items will be rated on 5-point Likert scale (1=never to 5=always) regarding the last 3–5 families taken care of by the FN. A low mean score indicates low self-perceived success in FSI performance whereas a higher score means higher succeed in intervention delivery. The Fidelity-SRT was pretested by two KP and requires approximately 10 min to complete.

#### Intervention Capacity Assessment

To assess FN's availability as defined per trial protocol to ensure provision of intervention capacity (fidelity delivery), the Intervention Capacity Assessment (ICA) will be completed once during process evaluation by KP's. The ICA consists of two self-developed items referring to the 4 weeks prior to the assessment and include: (1) was there always at least one FN available to deliver the FSI during 5 days per week?; (2) (only for FN) to what extent was it possible to provide the FSI to family members according to the timeline requirements of the protocol and the needs of the family? Both items will be rated on a 5-point Likert scale ranging from 1 (never) to 5 (always).

#### Intervention Fidelity Logs

FSI characteristics, such as duration in minutes, frequency, delivery mode, intervention activity (as defined in the intervention manual) will be recorded in a structured, online intervention log, which is part of the clinical trial electronic data capture system REDCap.[65] The logs are completed by FN for each included trial participant on an ongoing basis.

#### Family Nursing Practice Scale

To assess FN's appraisal of practice skills and their reflection in working with families, the 10-item German version of the 'Family Nursing Practice Scale (FNPS)' will be used.[66] Items will be scored on a 5-point Likert-type scale ranging from 1 (high level) to 5 (low level). Lower mean scores will represent higher practice skills. The FNPS was validated in critical care, suggesting high internal consistency (Cronbach's alpha 0.84).[66]

#### Feasibility Intervention Measure, Acceptability of Intervention Measure and Intervention Appropriateness Measure

To assess KPs' perceived level of feasibility, acceptability and appropriateness of the FSI, adapted psychometrically validated German versions of the Feasibility Intervention Measure (FIM), Acceptability of Intervention Measure (AIM) and Intervention Appropriateness Measure (IAM) will be completed two times. Each of these surveys include four items that use a 5-point Likert scale ranging from 1 (completely disagree) to 5 (completely agree). A recurring adaption is made by inserting 'FSI' into the appropriate space on the instrument for example, '(insert Intervention) is doable' will become 'FSI is doable'. All three surveys are reliable and pragmatic instruments with a high Crohnbach's alpha (FIM α=0.89; AIM α=0.80, IAM; α=0.87).[60 67]

#### Normalisation Measure Development questionnaire

To assess HP's perceived level of integration, the German version of the Normalisation Measure Development (NoMAD) will be used two times.[68] It includes 20 items using a 5-point Likert scale ranging from 1 (completely disagree) to 5 (completely agree) and is adaptable by specifying the '(the intervention)' to study purpose,[69] that is, the 'FSI'. The NoMAD instrument has high face validity, construct validity and internal consistency for assessing perceptions of factors relevant to the integration of the FSI. Internal consistency (Cronbach's alpha) was reported as follows: coherence (four items, α=0.71); collective action (seven items, α=0.78); cognitive participation (four items, α=0.81) and reflexive monitoring (five items, α=0.65). Overall, the normalisation scale was described as highly reliable (20 items, α=0.89).[69]

## Sustainability implementation survey

To assess both, KP's appraisal of sustainability determinants and post implementation sustainment outcomes, a self-developed sustainability survey will be sent at 6 and 12 months after the last intervention delivery and before the follow-up completion by the last study participant. The survey will include questions about the extent to which the FSI is still delivered as intended after FICUS study support is terminated (sustainability) and continued enactment of processes and practices in daily ICU routine that have been conveyed and learnt through the FSI.[70] All items will be rated on a 5-point Likert scale ranging from 1 (completely disagree) to 5 (completely agree).

## Demographics

Information on participants' age, gender, profession, institution, level of employment, level of education, work experience in the ICU, overall work experience in the profession and whether family nursing was part of the curriculum of their training and/or studies, will be collected. Furthermore, cluster data at ICU level will be collected to describe the eight cases in detail. Cluster data will include ICU cluster size (number of HP), number of beds, number of admissions per year and average length of stay.

All surveys will be implemented in UNIPARK software.

## Data analysis
### Qualitative data analysis

Data analysis of the process evaluation will require prompt analysis of interview data to rapidly develop or modify implementation strategies in case of need. Hence, small group interviews will be analysed using an adapted approach of Rapid Identification of Themes from Audio recordings,[71] in combination with a deductive approach guided by NPT.[47] This form of directed content analysis will apply the following steps: (1) specifying the evaluation foci, (2) identifying themes from audio recordings, (3) creating a codebook, (4) coding and refining the codebook and (5) sorting codes into themes inductively and deductively drawing on NPT constructs.[72]

Qualitative data from the summative evaluation (T4) will be analysed using content analysis, which allows the use of deductive and inductive strategies. It uses the following steps: (1) reading each of the transcribed interviews to get familiar with the data, taking notes of first impressions to gain a sense of the whole phenomenon, (2) identifying and developing codes for each unit of meaning, (3) developing of definitions for each code and group them into categories of related meanings and (4) synthesising codes and categories into themes.[72 73]

### Quantitative data analysis

Raw data will be verified, encrypted and entered into R Project for analysis. Descriptive statistics will be computed for the entire sample, by case and by data collection timepoint. Depending on scale level and distribution, results will be presented either as absolute and relative frequencies, as mean and SD, or as median and IQR. If scale levels and distributions allow, differences in central tendency will be tested by Student's t-test (paired or unpaired), with Welch's correction in the case of variance heterogeneity (unpaired case). Otherwise, differences in the distribution of ordinal data will be tested by Wilcoxon's signed-rank test (paired case within the same clusters or individuals over two timepoints, eg, Fidelity-SRT or NoMAD; unpaired case between clusters or individuals, eg, FNPS at baseline, referred to as Mann-Whitney U test). For nominal comparisons between clusters or individuals (unpaired), a $\chi^2$ test will be used if observed and expected frequencies allow for it, and Fisher's exact test will be used otherwise. For nominal comparisons within clusters or individuals over different timepoints (paired), McNemar's test will be used (or Cochran's Q test if a binary attribute is compared over more than two matched observations).[74]

### Mixed-methods and cross-case analysis

To provide an in-depth understanding of the phenomenon under study, and to increase the validity of our findings, data triangulation will occur on case level by comparing qualitative and quantitative data sets.[44 75] Data comparison occurs after the separate analysis of qualitative and quantitative data.[43 76 77] Datasets will be merged by listing and joining results on a same table, summarising raw data that are important to consider and displaying statistics and themes of each case in a matrix. Qualitative data will be mixed with quantitative data in relation to the implementation outcomes (ie, qualitative data on acceptability with IAM data) and the implementation process (ie, qualitative data on integration with NoMAD data) using comparing, contrasting and constantly verifying methods.[77]

Integration will also occur through construction of cases to develop a robust case presentation.[42 75] We will compare-contrast cases to not only identify patterns within but also across cases to build explanations for a successful implementation. To compare different cases, cross-case synthesis will be used, to compare and to identify similarities and differences between the cases. This will involve tables that display the data according to a uniform framework to compare across cases for both research questions and site-specific experiences to emerge from the data.[44] Finally, data will be brought together in a logic model including moderating influencers.[78]

## Data management

Intervention log data is stored on a MySQL database server hosted by the Clinical Trials Unit Zurich, which holds a REDCap End-User License Agreement for this electronic data capture system.[4] Unencrypted primary data, such as fieldnotes and audiotapes, will be stored in a lockable compartment to which only the research team has access. All documents are kept encrypted on the University of Zurich server and the key will be destroyed

after completion of the study. Essential documents will be retained for at least 10 years after termination of the study.

## Contribution to the literature and dissemination

This protocol outlines the rationale, design and methods for a process and outcome evaluation of a new, multicomponent, nurse-led FSI in ICUs. The key feature of the study design is the development of flexible and pragmatic data collection methods to capture data across five areas: (1) if the FSI was delivered as intended (fidelity), (2) how clinicians experienced the implementation process of the FSI, (3) which determinants (barriers and facilitators) influenced the integration process, (4) the impact of tailored implementation strategies to overcome contextual determinants and (5) to explain/interpret the clinical effectiveness results. Findings will be used to build a comprehensive understanding of how and why the FSI in ICUs was effectively implemented and delivered, and to delineate recommendations for scale-up to further ICUs and/or adaptions to different clinical contexts. Study findings will make a significant contribution to the current body of knowledge in attending to implementation of FSIs, as there is only currently one study protocol by Curtis et al[27] outlining plans to explore implementation. This study protocol may serve as a useful guidance for planning similar studies and projects that aim to test and/or implement complex interventions in real-world contexts, such as a nurse-led family support programme in ICUs.

Our findings will be disseminated to all partners and stakeholders involved and interested in its long-term implementation. Research findings will be disseminated via peer-reviewed journals and conferences.

## Ethics

The FICUS study, including the embedded implementation study has been approved by the responsible Swiss cantonal ethics committees (Nr. 202102300). Study participants will be informed about the purpose of this study and study participation orally and received a study information pack. Relevant national and international data protection regulation will be respected in accordance with the principles of the Declaration of Helsinki.[79]

**Contributors** SO, LV and RN conceptualised the study. The manuscript was written by SO and LV. RN provided ongoing feedback and critically revised the manuscript. MR supported the designing of the statistical analysis plan. TF and LC critically reviewed the study design and manuscript for important intellectual content. All authors have reviewed the drafts and approved the final version.

**Competing interests** None declared.

**Patient and public involvement** Patients and/or the public were not involved in the design, or conduct, or reporting, or dissemination plans of this research.

**Patient consent for publication** Not applicable.

**Provenance and peer review** Not commissioned; externally peer reviewed.

**ORCID iD**
Saskia Oesch http://orcid.org/0000-0001-6639-5875

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
