## [Reviewer comments · BMJ Open]

ARTICLE DETAILS

TITLE (PROVISIONAL)	Implementation of a multicomponent family support intervention in adult intensive care units – Study protocol for an embedded mixed methods multiple case study (FICUS Implementation Study)
AUTHORS	Oesch, Saskia; Verweij, Lotte; Clack, Lauren; Finch, Tracy; Riguzzi, Marco; Naef, Rahel

VERSION 1 – REVIEW

REVIEWER	Li Yoong Tang Universiti Malaya, NURSING SCIENCE
REVIEW RETURNED	28-May-2023

GENERAL COMMENTS	This is a comprehensive and good study for family in ICU setting. However, the interview questions for small groups and focus groups should be stated more specific (not just social and team processes) . Should state the statistics are used are they appropriate and described fully?
---

REVIEWER	Gianfranco Martucci Palliative Care Unit, Azienda USL - IRCSS Reggio Emilia, Italy
REVIEW RETURNED	30-May-2023

GENERAL COMMENTS	MINOR SUGGESTIONS TO THE AUTHORS:  - can't find the tag "figure 2" on the related item line 41: consider to delete "pragmatic" that is repeated twice; in generale, table 1 could be more visually appealing or easy to read, for example using bullet points 149: please specify if and how the data collection from KP is supervised or they are trained to do so. Generally, acquiring mixed or qualitative data requires some degree of training or at least supervision from an experienced researcher. - The theoretical models that you chose seem close to Pawson's realist theory, consider the hypothesis to cite it (if it's been part of the theoretical background of the protocol)? I don't think I saw anything on realist theory in your paper but I might be wrong. - I can't find the estimated number of small group interviews (line 190). Even if there is surely some flexibility, I think it's useful to have an initial hypothesis. In addition to this, maybe a short explanation of your definition of "small group interview" vs "focus group" might be useful to the reader. There are several ways of defining these things in qualitative research, so it might be useful to help the reader a bit. 229: Consider if a short description of what a "case conference" is in
--

	your model might be useful. line 241: there is a typo with a broken link. line 252: for what is my understanding, there is no such thing as a likert scale from 0 to “I don’t know”. Please correct the instrument or the sentence accordingly. 284: A comma after “outcomes” might improve readability 289: Small typo with the use of parentheses
--	--

VERSION 1 – AUTHOR RESPONSE

Reviewer 1	
The interview questions for small groups and focus groups should be stated more specific (not just social and team processes).	We have added more detailed information on interview questions for the small group interviews in line 198 and for the focus group interviews in line 224. At this stage, the detailed interview questions have not yet been finalized, but will then be presented in detail in the corresponding publication. Small group interviews will include questions such as: Was the defined implementation strategy useful, and do any adjustments or additional measures need to be taken? What are the reasons that currently hinder the implementation of the FSI, and what have been the biggest challenges so far and what are some ideas on how to overcome these barriers in the next phase of implementation? and, is the FSI capable of addressing the needs of the families and patients and how well does the FSI fit into current practices and processes, and does it require any adjustments? A semi-structured interview guide will be used including open-ended questions focusing on: How do ICU staff describe the implementation and feasibility of the FSI on their ICU, what team process-es and implementation activities took place during the implementation of the FSI? What factors were supportive or inhibiting and how did the team perceive the role of FN’s and the family care pathway? and what impact (benefits, unexpected consequences) did they experience in their daily work, interprofessional and with patients and their families?
Should state the statistics are used are they appropriate and described fully?	More details on the statistical procedure are inserted in the paragraph “quantitative analysis” in line 338. If scale levels and distributions allow, differences in central tendency will be tested by Student’s t-test (paired or unpaired), with Welch’s correction in the case of variance heterogeneity (unpaired case). Otherwise, differences in the distribution of ordinal data will be tested by Wilcoxon’s signed-rank test (paired case within the same clusters or individuals over two timepoints, e.g., Fidelity-SRT or NoMAD; unpaired case between clusters or individuals, e.g., FNPS at baseline, referred to as Mann & Whitney’s U-test). For nominal comparisons between clusters or individuals (unpaired), a chi-squared test will be used if

	observed and expected frequencies allow for it, and Fisher's exact test will be used otherwise. For nominal comparisons within clusters or individuals over different timepoints (paired), McNemar's test will be used (or Cochran's Q test if a binary attribute is compared over more than two matched observations).
Reviewer 2	
can't find the tag "figure 2" on the related item	Edited, tag for figure 2 is in line 142 and line 147
line 41: consider deleting "pragmatic" that is repeated twice in general, table 1 could be more visually appealing or easy to read, for example using bullet points	One "pragmatic" deleted as recommended I agree and inserted bullet points to improve readability as recommended in line 146
149: please specify if and how the data collection from KP is supervised or they are trained to do so. Generally, acquiring mixed or qualitative data requires some degree of training or at least supervision from an experienced researcher.	We have added a statement in line 154 The senior researcher is trained and skilled in qualitative data collection and will train and supervise the doctoral student in doing so.
The theoretical models that you chose seem close to Pawson's realist theory, consider the hypothesis to cite it (if it's been part of the theoretical background of the protocol)? I don't think I saw anything on realist theory in your paper, but I might be wrong.	Thank you very much for this important feedback. We didn't include realist theory as a methodological approach, but we incorporated a statement in the section of guiding conceptual frameworks and added a sentence including realist theory and references in line 142. This approach aligns with the realist evaluation paradigm because it attends to context, mechanisms, and outcomes of a complex intervention implementation, but uses specific theories and frameworks from implementation science to do so.
I can't find the estimated number of small group interviews (line 190). Even if there is surely some flexibility, I think it's useful to have an initial hypothesis. In addition to this, maybe a short explanation of your	We have added the number of small group interviews in line 199. A total of 16 small group interviews.. We have added the applied definition of small group interviews in line

definition of “small group interview” vs “focus group” might be useful to the reader. There are several ways of defining these things in qualitative research, so it might be useful to help the reader a bit.	196 and of focus group interviews line 215. Small group interviews refer to a semi-structured discussion with a small group of KP (4 to 5 participants) to facilitate effective discussion on the implementation process and ensure that each KP is adequately heard. This method will allow for gathering information, exploring opinions, and un-derstanding perspectives. Focus group interviews differ from small group interviews as they are more structured and con-ducted with a larger number of participants (8-10 participants) to obtain a comprehensive picture of the FSI implementation
229: Consider if a short description of what a “case conference” is in your model might be useful.	We have added a brief description of case conference in line 251 Monthly case conferences with FNs will be held as moderated group discussions to ensure quality of FSI delivery and to support consistency and adherence to the FSI activities. During case conferences, a specific family case will be presented by FN’s and FN’s will have the opportunity to discuss, to share their experiences and to reflect on the presented case. Based on the discussions, field notes will be taken by a doctoral student and FNs’ participation rate will be captured, as it is an indicator for fidelity receipt.
line 241: there is a typo with a broken link.	Edited and removed the broken link
line 252: for what is my understanding, there is no such thing as a likert scale from 0 to “I don’t know”. Please correct the instrument or the sentence accordingly.	Edited the instrument in line 277 Both items will be rated on a 5-point Likert scale ranging from 1 (never) to 5 (always).
284: A comma after “outcomes” might improve readability	Placed a comma
289: Small typo with the use of parentheses	Edited typo